# Comprehensive Molecular Profiling of Metastatic Pancreatic Adenocarcinomas

**DOI:** 10.3390/cancers17030335

**Published:** 2025-01-21

**Authors:** Vijay Antony, Tong Sun, Darin Dolezal, Guoping Cai

**Affiliations:** 1Department of Pathology, Yale University School of Medicine, New Haven, CT 06520, USA; vijay.antony@yale.edu (V.A.); tong.sun@yale.edu (T.S.); darin.dolezal@yale.edu (D.D.); 2Yale Cancer Center, Yale University School of Medicine, New Haven, CT 06520, USA

**Keywords:** pancreatic adenocarcinoma, metastasis, molecular profile, gene mutation, TCGA

## Abstract

Pancreatic ductal adenocarcinoma (PDAC) is a disease with a dismal prognosis and limited curative treatment options. Targeted molecular therapies may offer a ray of hope for these patients; however, there is a significant knowledge gap regarding the differences between primary PDAC and metastatic PDAC at the molecular level. Awareness of this critical information may help direct the development of personalized therapies for these patients. This study aimed to investigate comprehensive molecular profiles of metastatic PDAC in comparison to primary PDAC reported in our prior study and supplemented by TCGA data.

## 1. Introduction

Pancreatic ductal adenocarcinoma (PDAC) is notorious for its insidious nature, with about 80% of cases eluding detection until advanced stages. As one of the cancers with a worse prognosis, pancreatic carcinoma accounts for about 3% of all cancers but 8% of all cancer-related deaths in the US [1]. In contrast to the general trend of improved cancer survival over the past few decades, the 5-year relative survival rate for pancreatic cancer remains steady at about 12%, the lowest among all cancer types. This may most likely be related to the fact that most pancreatic cancers are diagnosed at an advanced stage. The available treatment modalities include surgery, radiation therapy, and chemotherapy. Still, they are seldom curative as the patients usually present at an advanced stage, which precludes them from being a surgical candidate.

The genomic landscape of pancreatic ductal adenocarcinoma (PDAC) has been defined by the four most commonly mutated pathways, including Kirsten rat sarcoma virus (*KRAS*) mutations, cellular tumor antigen p53 (*TP53*) mutations, cyclin-dependent kinase inhibitor *2A (CDKN2A*) alterations, and SMAD family member 4 (*SMAD4*) inactivation, which were called the “big four” [2,3]. DNA repair pathway deficiencies, notch pathway dysregulation, PI3K/AKT pathway activation, and other pathways were also proved to play roles in PDAC tumorigenesis and progress [4,5,6]. However, the difference in molecular features between primary PDAC and distant metastatic PDAC has not been fully explored [7]. Further understanding of the unique features of metastatic pancreatic carcinoma is essential for developing targeted therapies and improving advanced-stage pancreatic cancer patient outcomes [8,9,10,11,12,13,14]. Here, in this study, we focus on comprehensive molecular profiling of metastatic PDACs and subsequent comparison with the profiles of primary PDACs published in our prior work [15] as well as documented in the TCGA database.

## 2. Materials and Methods

### 2.1. Case Selection

With approval by the Institutional Review Board, the pathology database was searched for metastatic adenocarcinomas of pancreaticobiliary origin in which Oncomine Comprehensive Assay (OCA, ThermoFisher Scientific, Frederick, MD, USA) was performed at Yale New-Haven Hospital between 2015 and 2023. The patients’ demographic information, pathologic diagnosis, and molecular testing results were reviewed and analyzed. The timing between the appearance of metastases and the primaries was calculated using the mean of the time difference between the pathology reports of the primary diagnosis of the pancreatic adenocarcinoma and the subsequent confirmatory report of the metastasis of a non-contiguous site in each individual metastatic PDAC case. Additionally, for the TCGA pancreatic adenocarcinoma cohorts, only data from specimens classified as “primary tumor” were included in the analyses (N = 772); samples labeled as “metastasis” or with unspecified origins were excluded.

### 2.2. Pathology Specimen Processing

Specimen types submitted for molecular testing included biopsy, cytology, and resection specimens. The biopsy and cytology were performed via a percutaneous approach under CT imaging or endoscopic ultrasound guidance, depending on the location of the lesion. Plural fluid and peritoneal fluid were obtained by thoracentesis or paracentesis, respectively. The cytology specimens were fixed in Cyto-Rich fixative to prepare a cell block using the HistoGel technique (StatLab, McKinney, TX, USA). The biopsy and resection specimens were fixed in 10% formalin. Subsequently, cell block material or formalin-fixed tissue was processed through routine steps of formalin tissue fixation, dehydration, and paraffin embedding. A section of the formalin-fixed paraffin-embedded (FFPE) tissue or cell block was stained with Hematoxylin and Eosin (H&E) for microscopic pathologic evaluation.

### 2.3. Tumor Molecular Testing

Molecular profiling was performed in the Yale New-Haven Hospital Tumor Profiling Laboratory. Tumor cellularity was assessed by a pathologist. Samples with tumor cellularity of less than 10% were deemed insufficient for analysis. Manual or laser-captured microdissection was performed to enrich tumor cells in borderline cases. Matched germline control samples were collected from blood, buccal swabs, or non-tumor FFPE tissue. DNA/RNA extraction was performed using commercially available kits (Qiagen, Inc., RedWood City, CA, USA or Maxwell RSC by Promega, Madison, WI, USA), and nucleic acids were quantitated on a Qubit 2.0 fluorimeter. The targeted amplicon-based sequencing using three consecutive versions of the OCA (v1, v2, or v3) was conducted on an Ion Torrent platform. Each OCA version assessed tumor DNA for mutations and/or amplifications in 134, 134, and 146 cancer-related genes, respectively. Additionally, it analyzed tumor RNA for gene fusions involving 22, 23, and 44 oncogenic driver genes, respectively. The sequencing datasets underwent processing through an internally developed bioinformatic pipeline. The interpretation and clinical reporting of the results were carried out by subspecialty board-certified molecular pathologists.

### 2.4. Statistical Analysis

For statistical analysis, continuous data were presented as means with standard deviation (SD), and differences among fine-needle aspiration (FNA), fine-needle biopsy (FNB), and resection samples were assessed using analysis of variance. A post hoc Tukey honestly significant difference test was subsequently conducted to examine pairwise comparisons. Categorical data were represented by frequencies and percentages, and differences were analyzed using the Fisher exact test. Missing data were imputed as medians for numerical variables and labeled as “missing data” for categorical variables. Statistical significance was considered at a (2-tailed) *p*-value < 0.05. All statistical analyses were performed using GraphPad Prism 10 for macOS, version 10.1.1.

## 3. Results

### 3.1. Clinicopathological Characteristics of Metastatic Pancreatic Adenocarcinoma

A total of 115 metastatic PDAC cases were identified, which were from 64 (56%) men and 51 (44%) women with a mean age of 67 years (ranging from 36 to 91 years) (Table 1). The interval between the diagnosis of primary tumor and metastasis ranged from 0 to 143 months (mean = 11 months). The cohort cases included 13 FNA, 92 biopsy, and 10 resection specimens from disparate metastatic sites, including the liver (n = 77, 67%), lymph nodes (n = 11, 10%), omentum/peritoneum (n = 8, 7%), lung (n = 7, 6%), soft tissue (n = 6, 5%), biliary system (n = 2, 2%), bone (n = 1, 1%), duodenum (n = 1, 1%), ascitic fluid (n = 1, 1%), and pleural fluid (n = 1, 1%).

### 3.2. Specimen Adequacy for Molecular Testing

After the slide review, a total of 71 out of 115 cases (62%) were tentatively found to have adequate tumor cellularity for molecular testing (Table 2). The remaining 44 cases were found to have insufficient material, of which cytology cases were mostly inadequate for the test (*p* < 0.01). Microdissection, including both manual and laser capture, was further performed in 58 of 71 (82%) specimens (Table 2). The overall tumor cell percentage obtained post-microdissection was 49% on average, similar across the specimen types (*p* > 0.05). All 71 cases succeeded in DNA-based molecular testing, including analysis for gene mutations and copy number alterations, while 7 out of 71 cases failed in RNA-based gene fusion assay.

### 3.3. Molecular Profiles of Metastatic Pancreatic Adenocarcinoma

The cases of metastatic pancreatic adenocarcinoma showed a mean of 3.4 molecular alterations (ranging from 0 to 9 alterations per case) with a total of 239 molecular alterations (Table 3). The molecular alterations detected were mainly gene mutations (89.5%), with a small number (10.5%) showing gene copy number alteration. No gene fusions were identified in any cases of the cohort. A total of 35 genes showed mutations in one or more of the cases in this cohort (Figure 1). The most common mutated gene was *KRAS*, identified in 61 of 71 cases (86%), followed by *TP53* mutation, seen in 59 (83%) cases (Table 3). Other common mutated genes included *CDKN2A* and *ARID1A*, detected in 21% and 15% of cases, respectively (Figure 1). Of the *KRAS* mutations, the most common type was G12D, accounting for 44.3% of all *KRAS* mutations, followed by G12V (24.6%) and G12R (21.3%) mutations. In this current cohort, germline control specimens were available in 69 out of 71 (97%) cases, which yielded nine germline mutations that involved *BRCA2* (n = 3), *BRCA1* (n = 3), *TP53* (n = 1), *ATM* (n = 1), and *FANCA* (n = 1) genes, respectively.

Gene copy number alterations were identified in 19 cases involving 15 separate genes. Of the genes with copy number alterations, amplifications and deletions were seen in 13 and 2 genes, respectively. The most common genes showing amplification were *CCNE1* (n = 6) and *ERBB2* (n = 5), followed by *MYC* (n = 3) and *FGFR3* (n = 2) (Figure 1); other amplified genes included *PPARG*, *CCND3*, *CDK2*, *CDK4*, *KRAS*, *MET*, *AKT2*, *CDK6*, and *FLT3* (n = 1, each). The case that showed *KRAS* amplification also had a mutation in the *KRAS* gene. Gene deletions involving *CDKN2A* and *CDKN2B* genes were identified in a single case each.

### 3.4. Molecular Alterations in Primary vs. Metastatic Pancreatic Adenocarcinoma

We further compared our metastatic pancreatic adenocarcinoma cohort results with primary cancer results, including public TCGA database pancreatic adenocarcinoma cohort (N = 772) and our prior primary pancreatic adenocarcinoma cohort (N = 61) (Table 4). Notably, *KRAS* mutation was highly prevalent across all cohorts, with 92% in the TCGA cohort, 90% in our primary cohort, and 85.9% in the current metastatic cohort, with no statistically significant difference between the three cohorts (*p* > 0.05). Intriguingly, *TP53* mutation also demonstrated a substantial presence, with 64% in the TCGA cohort and 64% in our primary cohort, and notably increased to 83% in the current metastatic cohort (*p* = 0.001). Interestingly, *SMAD4* mutation seemed to be less likely to be seen in the current metastatic cohort than in the TCGA data or our primary cohort (*p* < 0.05). There were no differences in *CDKN2A* gene alterations across the different cohorts (*p* > 0.05). In addition, *ARID1A*, *BRAF*, and *PIK3CA* mutations seemed to increase in the metastatic cohort, compared with TCGA and in-house primary cancer cohorts (all *p* < 0.05).

## 4. Discussion

Tumorigenesis in PDAC is believed to follow a stepwise progression, starting from an initial low-grade pancreatic intraepithelial neoplasia (PanIN), then a high-grade PanIN, and eventually becoming invasive carcinoma [7]. In parallel, characteristic molecular alterations accumulate over the course of tumor progression [2,4,7,16,17,18,19,20]. *KRAS* gene mutations are the most documented molecular alterations in PDAC [2,3]. It is, however, believed to be one of the early molecular events during tumorigenesis and is identified in low-grade PanIN and even in histologically normal ductal epithelium [7]. *KRAS* mutations lead to constitutive activation of the RAS protein, resulting in activation of the RAF/MEK/ERK pathway or PI3K/PDK1/AKT/mTOR pathway and, ultimately, the development of intraepithelial neoplasia [21]. At the molecular level, high-grade PanIN is characterized by the occurrence of additional molecular alterations, primarily in tumor suppressor genes such as *TP53*, *CDKN2A*, and *SMAD4*, which impair DNA damage repair, inhibit apoptosis or override cell cycle checkpoints [7,22]. Initiation and subsequent progression of metastasis were initially hypothesized to be associated with molecular alterations in a set of genes different from the tumorigenesis genes mentioned earlier [8,22]. However, previous studies have shown no significant differences between the frequencies of specific genes affected in the primary PDAC tumors and metastatic PDAC tumors, even in matched individuals [22,23,24].

The outcomes of the present study mostly conform with the existing body of knowledge. In descending order, the most frequently mutated genes in the metastatic cohort are *KRAS*, *TP53*, and *CDKN2A*, found in 61 of 71 (86%), 59 of 71 (83%), and 15 of 71 (21%) cases, respectively. *ARID1A* mutations, affecting 11 of 71(15%) cases, emerged as the fourth most common, surpassing *SMAD4* mutations involving 7 of 71 (10%) cases, compared to the preceding study. The most common *KRAS* mutations in the current study, in the order of frequency, were G12D (n = 27/71, 38%), G12V (n = 15/71, 21%), and G12R (n = 13/71, 18%), which exactly mirrored the previous study [25]. Other genes found in this cohort mutated in lower frequencies include oncogenes such as *PIK3CA*, *PDGFRB*, *GNAQ*, *SF3B1*, *U2AF1*, *BRAF*, *TERT*, *GATA2*, *CREBBP*, *ROS1*, and *GNAS*, and tumor suppressor genes such as *ATRX*, *TSC1*, *BRCA1*, *BRCA2*, *POLE*, *SMARCA4*, *NF1*, *ATM*, *RNF43*, *FANCA*, *NBN*, *BAP1*, *RAD51B*, *NF2*, *MSH2*, and *STK11.*

Gene amplifications were detected in 18 cases of the metastatic PDAC cohort (n = 71) compared to 6 cases of the primary PDAC cohort (n = 61) (*p* = 0.0245). In the metastatic PDAC cohort, 13 genes out of 35 mutated genes were affected by gene amplifications, the most common one being *CCNE1* (six cases), compared to 8 genes out of 42 mutated genes in the primary PDAC cohort, with the most common ones being *MYC* (two cases) (*p* > 0.05). *CCNE1* amplification has been found to be associated with worse prognosis in a variety of metastatic cancers, including PDAC [26]. A recent copy number variation study showed that tumor fraction was remarkably higher in metastatic than localized tumors, and genetic heterogeneity was found between distinct metastatic tumors, especially in different organs [24].

The purpose of comprehensive molecular profiling as part of anticancer therapy is to identify actionable/druggable targets for treatment. As per NCCN Guidelines for pancreatic adenocarcinoma version 3.2024, potentially actionable targets for PDAC include fusions (*ALK*, *NRG1*, *NTRK*, *ROS1*, *FGFR2*, and *RET*), mutations (*BRAF*, *BRCA1/2*, *KRAS*, and *PALB2*), amplifications (*HER2*), microsatellite instability (MSI), mismatch repair deficiency (dMMR), or tumor mutational burden (TMB) via an FDA-approved and/or validated NGS-based assay [27]. In this regard, in addition to the previously mentioned *KRAS* gene mutations, the current cohort demonstrates mutations in *BRAF* (n = 5/71, 7.0%), *BRCA1* (n = 3/71, 4.2%), and *BRCA2* (n = 6/71, 8.5%), *HER2* amplifications (n = 5/71, 7%), and one MMR gene (*MSH2)* mutation (n = 1/71, 1.4%). Recommended first-line targeted molecular therapy regimens for locally advanced disease and metastatic disease include Entrectinib/Larotrectinib (if *NTRK* gene fusion-positive), Pembrolizumab (if MSI-high, dMMR, or TMB-high [≥10 mut/Mb]), Dabrafenib + trametinib (if *BRAF* V600E mutation-positive), and Selpercatinib (if *RET* gene fusion-positive). Recommended maintenance targeted molecular therapy regimens for metastatic disease include Olaparib, a poly (ADP ribose) polymerase (PARP) inhibitor (only for patients who have had prior platinum-based chemotherapy and carry germline *BRCA*1/2 mutations), and Rucaparib (for patients who have had prior platinum-based chemotherapy and carry germline or somatic *BRCA*1/2 mutations or *PALB2* mutations). Preferred subsequent targeted molecular therapy regimens for locally advanced/metastatic disease and recurrent disease include Entrectinib/Larotrectinib (if *NTRK* gene fusion-positive) and Pembrolizumab (if the patient has not received any prior immunotherapy and is MSI-high, dMMR, or TMB-high [≥10 mut/Mb]). Other recommended targeted molecular regimens for locally advanced/metastatic and recurrent diseases include Dabrafenib + trametinib (if *BRAF* V600E mutation-positive) and Selpercatinib (if *RET* gene fusion-positive). If the patient has not received prior immunotherapy, Dostarlimab-gxly can be used (if MSI-high or dMMR), or a combination of Nivolumab + Ipilimumab can be used (if TMB-high [≥10 mut/Mb]). Additionally, as a part of subsequent therapy for locally advanced/metastatic disease or if the patient has recurrent disease, Adagrasib or Sotorasib may be used (if the patient is positive for *KRAS* G12C mutation) [27,28].

A few intriguing findings were revealed when comparing the TCGA molecular profile and in-house primary pancreatic adenocarcinoma with the current metastatic cohort. First, in metastatic carcinoma, there was a notable escalation in the frequency of *TP53* gene mutations. As a critical tumor suppressor gene, the mutated p53 protein contributes to uncontrolled cell growth, evasion of apoptosis, and increased genomic instability. This altered molecular landscape fosters a more aggressive phenotype, promoting the invasion and dissemination of cancer cells to distant sites [2,29,30]. Second, an increased prevalence of *ARID1A* mutations was noted in the pancreatic adenocarcinoma. As a key component of the SWI/SNF chromatin remodeling complex, the mutation of the *ARID1A* gene is associated with chromatin remodeling dysfunction, leading to alteration in gene expression and cellular processes [2,31]. The loss of *ARID1A* function appears to contribute to enhanced invasiveness and metastatic protein by promoting changes in the tumor microenvironment and facilitating the acquisition of aggressive traits. Similarly, another critical component of the SWI/SNF chromatin remodeling complex, loss of *SMARCA4* function, has been linked to increased genomic instability, altered cell differentiation, and enhanced tumor invasiveness [32,33]. The current study showed an increased alteration rate of *SMARCA4* mutation in the metastatic pancreatic adenocarcinoma cohort compared with the primary PDAC cohorts, though statistical significance has not been reached. Third, oncogenes of *PIK3CA* and *BRAF*, which are critical components of PI3K/AKR/mTOR and RAS/RAF/MEK/ERK signaling pathways [2], showed increased mutations in the metastatic pancreatic adenocarcinoma cohort. Activation of these pathways can contribute to increased invasiveness, angiogenesis, and evasion of apoptosis, all of which are critical aspects of metastatic progression and potential therapeutic targets [34,35]. Finally, there seemed more metastatic carcinomas carried *BRCA2* mutation as compared to the primary tumors, which suggests an increasing role of PARP inhibitors in the treatment of metastatic pancreatic cancer [27,36,37]. Thus, comprehensive molecular profiling, using clinically validated testing platforms, of metastatic PDAC may assist in finding potential targets, leading to more personalized treatment options for patients with PDAC.

There are several drawbacks to the current study. First, this was a small-scale retrospective study, which may prompt selection bias. Secondly, the comparison of molecular alterations between metastatic and primary PDAC was based on unmatched pairs of tumors. These two cohorts, however, shared similar characteristics, including patients’ age distribution, specimen types, sample processing, and molecular testing platforms. Although we found several pathogenic alterations with increased frequencies in metastatic PDAC, it is still unclear whether these alterations are related to tumor progression and metastasis or whether the tumors with these molecular alterations are prone to metastasis. Third, three different OCA versions were used in the current study cohort, owing to assay upgradation between 2015 and 2023. Although the numbers of genes tested across different versions were not hugely different, the detection of specific gene alterations may be compromised due to inadequate coverage of genes in the test. Lastly, OCA is not an ideal platform to identify gene copy number variations since only limited genes were tested. Nevertheless, OCA is the most common molecular test routinely performed for pathologic samples, the results of which have significant therapeutic implications. Comprehensive genomic changes associated with metastasis will contribute to a better understanding of the disease and guide the development of more effective therapeutic strategies for patients with metastatic PDAC.

## 5. Conclusions

Comparison of the current metastatic PDAC cohort and the unmatched primary PDAC cohort of Razzano et al., supplemented with the public PDAC TCGA data, revealed increased *TP53*, *ARID1A*, *BRCA2*, *BRAF*, and *PIK3CA* mutations in metastatic cases. These findings suggest metastatic PDAC may differ from primary PDAC, offering potential therapeutic targets.

## Figures and Tables

**Figure 1 cancers-17-00335-f001:**
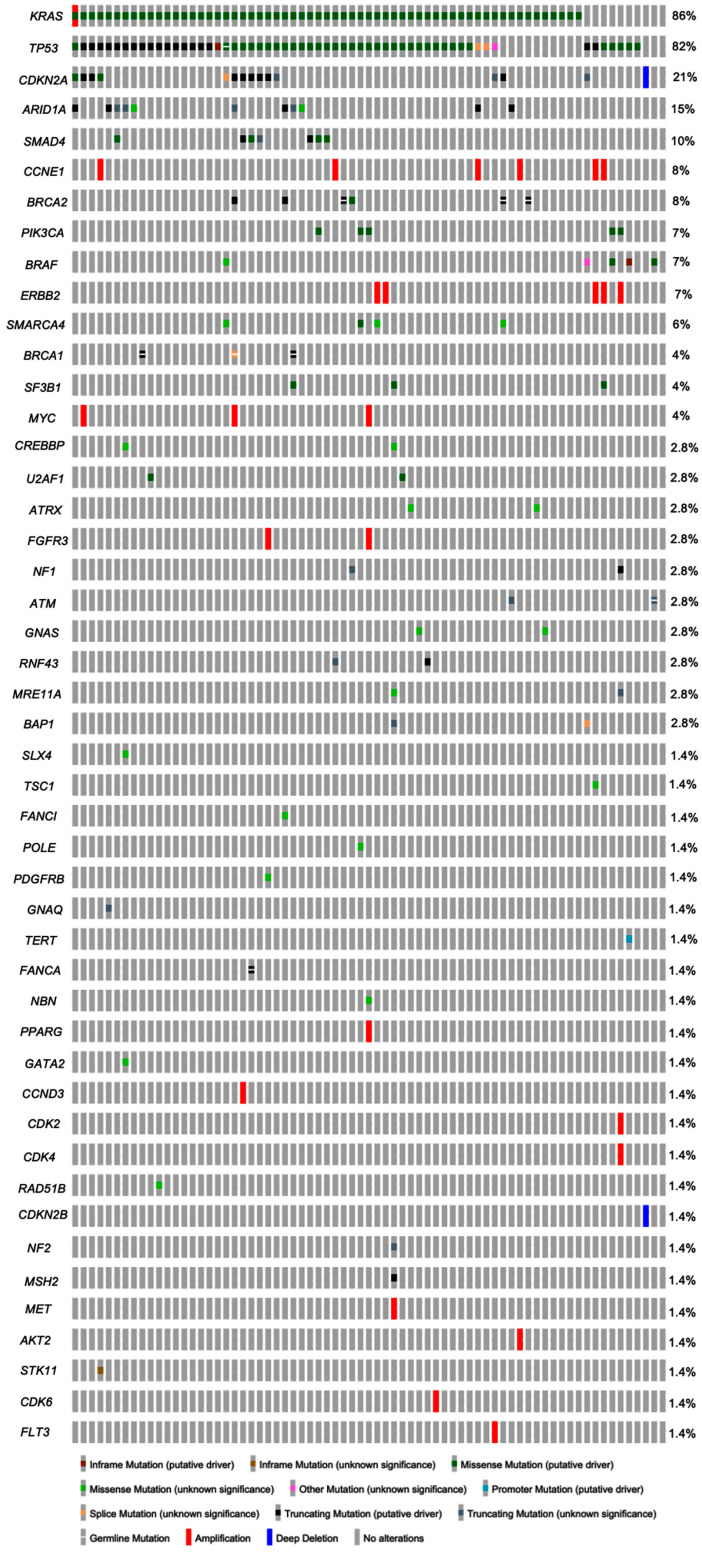
Molecular alterations in metastatic pancreatic adenocarcinoma.

**Table 1 cancers-17-00335-t001:** Clinicopathologic features of metastatic pancreatic adenocarcinoma cohort.

Age, Years	
Mean	67
Range	36–91
Gender, n (%)	
Male	64 (56%)
Female	51 (44%)
Interval between primary and metastasis, months	
Mean	11
Range	0–143
Specimen type, n (%)	
Core biopsy	92 (80%)
Cytology	13 (11%)
Resection	10 (9%)
Metastatic site, n (%)	
Liver	77 (67%)
Lymph node	11 (10%)
Omentum/peritoneum	8 (7%)
Lung	7 (6%)
Soft tissue	6 (5%)
Biliary tract	2 (2%)
Bone	1 (1%)
Duodenum	1 (1%)
Ascitic fluid	1 (1%)
Pleural fluid	1 (1%)

**Table 2 cancers-17-00335-t002:** Adequacy assessment for molecular testing by specimen types.

	Total (n = 115)	Biopsy (n = 92)	Cytology (n = 13)	Resection (n = 10)	*p*
Microdissection, n (%)					0.0085 *
Yes	58 (51%)	49 (53%)	1 (8%)	8 (80%)	
No	13 (11%)	11 (12%)	1 (8%)	1 (10%)	
Inadequate	44 (38%)	32 (35%)	11 (84%)	1 (10%)	
Tumor proportion (%)					0.6613
Mean (SD)	49 (20)	50 (20)	45 (7)	41 (24)	
Range	10–90	15–90	40–50	10–75	
DNA analysis, n (%)					0.0012 *
Yes	71 (62%)	60 (65%)	2 (15%)	9 (90%)	
No	44 (38%)	32 (34%)	11 (85%)	1 (10%)	
RNA analysis, n (%)					0.0175 *
Yes	64 (56%)	55 (60%)	2 (15%)	7 (70%)	
No	51 (44%)	37 (40%)	11 (85%)	3 (30%)	

* *p* < 0.05 by analysis of variances followed by post hoc Turkey test.

**Table 3 cancers-17-00335-t003:** Molecular alterations of metastatic pancreatic adenocarcinomas (n = 71).

Molecular Alterations, per Case	
Mean	3.4
Range	0–9
Type of molecular alterations, n (%)	239 (total)
Gene mutation	214 (89.5)
Gene copy number alteration	25 (10.5)
Gene fusion	0 (0)
Common gene mutations, n (%) *	
*KRAS*	61 (85.9)
*TP53 ***	59 (83.1)
*CDKN2A*	15 (21.1)
*ARID1A*	11 (15.5)
*SMAD4*	7 (9.9)
*BRCA2 ****	6 (8.5)
*PIK3CA*	5 (7.0)
*BRAF*	5 (7.0)
*SMARCA4*	4 (5.6)
*SF3B1*	3 (4.2)
*BRCA1 *****	3 (4.2)
*KRAS* mutations, n (%)	61 (total)
G12D	27 (44.3)
G12V	15 (24.6)
G12R	13 (21.3)
G12C	1 (1.6)
G12S	1 (1.6)
G13D	1 (1.6)
Q61H	1 (1.6)
Q61R	1 (1.6)
Q61L	1 (1.6)

* Germline mutations were identified in 9 out of 71 cases (12.7%), including 3 *BRCA1*, 3 *BRCA2*, 1 *TP53*, 1 *ATM*, and 1 *FANCA* gene mutation; ** 1 out 59 *TP53* mutations was germline mutation; *** 3 out of 6 *BRCA2* mutations were germline mutations; **** 3 out of 3 *BRCA1* mutations were germline mutations.

**Table 4 cancers-17-00335-t004:** Comparison of molecular profiles in primary and metastatic pancreatic adenocarcinoma.

Common Mutated Genes	TCGA Cohort (N = 772)	Primary Cohort * (N = 61)	Metastatic Cohort (N = 71)	*p*-Values
*KRAS*	709/772 (91.8%)	55/61 (90.2%)	61/71 (85.9%)	>0.05
*TP53*	495/772 (64.1%)	39/61 (63.9%)	59/71 (83.1%)	0.003
*SMAD4*	163/772 (21.1%)	9/61 (14.8%)	7/71 (9.9%)	0.041
*CDKN2A*	124/772 (16.1%)	15/61 (24.6%)	15/71 (21.1%)	>0.05
*ARID1A*	49/772 (6.3%)	2/61 (3.3%)	11/71 (15.5%)	0.013
*BRCA2*	11/772 (1.4%)	4/61(6.6%)	6/71 (8.5%)	0.0003
*PIK3CA*	12/772 (1.6%)	1/61 (1.6%)	5/71 (7.0%)	0.0196
*BRAF*	7/772 (0.9%)	1/61 (1.6%)	5/71 (7.0%)	0.0026
*SMARCA4*	16/772 (2.1%)	1/61 (1.6%)	4/71 (5.6%)	>0.05
*SF3B1*	16/772 (2.1%)	0/61 (0)	3/71 (4.2%)	>0.05
*BRCA1*	8/772 (1.0%)	0/61 (0)	3/71 (4.2%)	>0.05

* Data were derived from the previously published article [15].

## Data Availability

The data presented in this study are available on request from the corresponding author.

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
