# Peer review of "Comprehensive Molecular Profiling of Metastatic Pancreatic Adenocarcinomas"

_cancers, 2025, doi:10.3390/cancers17030335_

Round 1
Reviewer 1 Report
Comments and Suggestions for Authors
Since even in the era of the molecularly targeted therapies the treatment options of the metastatic pancreatic cancer are rather modest, the more profound understending of the molecular backgrounds is of pivotal importance. The carefully designed and presented manuscript does reinforce that the metastatic tumors differ from the primary ones, might involving alterate treatment modalities.
The primary and the metastatic cases are unmatched, but comparable. However, the original Razzano's paper and the present Table 4. contain different data (SMAD 4: 9/61 - 14.8%, but here: 10/61 - 16.4 %; BRCA2: Fig.. 2: 4 cases, but here: 5/61).
The numbers in the Table 4 and those in the Discussion section are not identical (59/71; 83,1%, but in line 193: 61/71; 86%. Similarly: CDKN2A: 15/71; 21.1%, vs. line 193: 14/71; TP53: 59/71 in the Table, while in line 193: 58/71). The discrepancies should be clarified.
Author Response
Reviewer #1
Since even in the era of the molecularly targeted therapies the treatment options of the metastatic pancreatic cancer are rather modest, the more profound understanding of the molecular backgrounds is of pivotal importance. The carefully designed and presented manuscript does reinforce that the metastatic tumors differ from the primary ones, might involving alternate treatment modalities.
The primary and the metastatic cases are unmatched, but comparable. However, the original Razzano's paper and the present Table 4. contain different data (SMAD 4: 9/61 - 14.8%, but here: 10/61 - 16.4 %; BRCA2: Fig. 2: 4 cases, but here: 5/61).
Response: These were typographical errors which we have corrected to match with the Razzano paper. Thank you for pointing it out.
The numbers in the Table 4 and those in the Discussion section are not identical (59/71; 83,1%, but in line 193: 61/71; 86%.
Response: The 59/71 (83.1%) in Table 4 was referring to the frequency of TP53 mutations in the metastatic cohort while the 61 of 71 (86%) was referring to the frequency of KRAS mutations in the metastatic cohort.
Similarly: CDKN2A: 15/71; 21.1%, vs. line 193: 14/71; TP53: 59/71 in the Table, while in line 193: 58/71). The discrepancies should be clarified.
Responses: These were typographical errors which we have corrected in the body of discussion.
Thank you for pointing it out.

Reviewer 2 Report
Comments and Suggestions for Authors
This well-written manuscript suggests metastatic PDAC possesses unique genetic characteristics, including increased rates of TP53, ARID1A, BRAF, and PIK3CA mutations in metastatic diseases compared to primary PDAC. I have minor comments.
1- The abbreviations should be expanded at the first mention e.g., FNA, FNB, etc.
2-In Table 3 and its legend, please revise the abbreviation BRCA1, it is sometimes written BRAC1.
3- The resolution of Figure 1 is too little, I can't clearly read the figure.
4- Do the primary PDAC samples and TCGA cohorts used in comparison contain metastatic and non-metastatic cases? Please define these cases in the method section.
Author Response
Reviewer #2
This well-written manuscript suggests metastatic PDAC possesses unique genetic characteristics, including increased rates of TP53, ARID1A, BRAF, and PIK3CA mutations in metastatic diseases compared to primary PDAC. I have minor comments.
- The abbreviations should be expanded at the first mention e.g., FNA, FNB, etc.
Response: These has been spelled out in the revised manuscript.
- In Table 3 and its legend, please revise the abbreviation BRCA1, it is sometimes written BRAC1.
Response: All BRACs have been corrected to BRCAs. Thank you for pointing it out.
- The resolution of Figure 1 is too little, I can't clearly read the figure.
Response: We have modified the Figure 1 with image enhancement technique, resulting in a better contrast.
4- Do the primary PDAC samples and TCGA cohorts used in comparison contain metastatic and non-metastatic cases? Please define these cases in the method section.
Response: For the TCGA pancreatic adenocarcinoma cohorts, only data from specimens classified as “primary tumor” were included in the analyses (N = 772). Samples labeled as “metastasis” or with unspecified origins were excluded. Additional description has been added to the Methods section (Page 2, lines 67-70). In our own cohort of primary PDAC, molecular test was performed on primary not metastatic tumor as detailed in Razzano’s paper.

Reviewer 3 Report
Comments and Suggestions for Authors
In this manuscript, the authors described the differences of the molecular genetic profiling between primary and metastatic pancreatic adenocarcinoma using OCA. Overall research design is concreate and the article is well written. Although there are some reports about the moecular profiling of metastatic pancreatic adenocarcinoam using OCA, this study has some degree of novelty because true somatic mutations were confirmed by the paired normal samples whatever the tissue type is (blood or FFPE). However, it did not provide sufficient information whether data obtained from primary and metastatic samples were directly comparable. Although the possible influence of OCA versions on the results was describe in the discussion section, more comprehensive description about the QC is required to compare the primary and metastatic disease.
Author Response
Reviewer #3
In this manuscript, the authors described the differences of the molecular genetic profiling between primary and metastatic pancreatic adenocarcinoma using OCA. Overall research design is concreate and the article is well written. Although there are some reports about the molecular profiling of metastatic pancreatic adenocarcinoma using OCA, this study has some degree of novelty because true somatic mutations were confirmed by the paired normal samples whatever the tissue type is (blood or FFPE). However, it did not provide sufficient information whether data obtained from primary and metastatic samples were directly comparable.
Response: As stated in our manuscript, the comparison of molecular profiles between primary and metastatic PDAC was analyzed on unmatched samples. This is one of drawbacks of the current study. Even though, we tried to do our best to make these two cohorts as comparable as possible. These two cohorts have similar patients’ demographics and study periods, similar specimen processing, and identical testing platforms.
Although the possible influence of OCA versions on the results was described in the discussion section, more comprehensive description about the QC is required to compare the primary and metastatic disease.
Response: Additional description of the QC required to ensure comparability between the primary PDAC (Razzano) cohort and metastatic cohort (current study) has been added to the Discussion section (Page 10, lines 202-210).
